# PixelSNAIL: An Improved Autoregressive Generative Model

**Xi Chen, Nikhil Mishra, Mostafa Rohaninejad, Pieter Abbeel**
Embodied Intelligence
UC Berkeley, Department of Electrical Engineering and Computer Sciences

## Abstract

Autoregressive generative models achieve the best results in density estimation tasks involving high dimensional data, such as images or audio. They pose density estimation as a sequence modeling task, where a recurrent neural network (RNN) models the conditional distribution over the next element conditioned on all previous elements. In this paradigm, the bottleneck is the extent to which the RNN can model long-range dependencies, and the most successful approaches rely on causal convolutions. Taking inspiration from recent work in meta reinforcement learning, where dealing with long-range dependencies is also essential, we introduce a new generative model architecture that combines causal convolutions with self attention. In this paper, we describe the resulting model and present state-of-the-art log-likelihood results on heavily benchmarked datasets: CIFAR-10 (2.85 bits per dim), $32 \times 32$ ImageNet (3.80 bits per dim) and $64 \times 64$ ImageNet (3.52 bits per dim). Our implementation is available at https://github.com/neocxi/pixelsnail-public.

## 1 Introduction

Autoregressive generative models over high-dimensional data $\mathbf{x} = (x_1, \ldots, x_n)$ factor the joint distribution as a product of conditionals:

$$p(\mathbf{x}) = p(x_1, \ldots, x_n) = \prod_{i=1}^{n} p(x_i | x_1, \ldots, x_{i-1})$$

A recurrent neural network (RNN) is then trained to model $p(x_i | x_{1:i-1})$. Optionally, the model can be conditioned on additional global information $\mathbf{h}$ (such as a class label, when applied to images), in which case it in models $p(x_i | x_{1:i-1}, \mathbf{h})$. Such methods are highly expressive and allow modeling complex dependencies. Compared to GANs Goodfellow et al. (2014), neural autoregressive models offer tractable likelihood computation and ease of training, and have been shown to outperform latent variable models van den Oord et al. (2016b;a); Salimans et al. (2017).

The main design consideration is the neural network architecture used to implement the RNN, as it must be able to easily refer to earlier parts of the sequence. A number of possibilities exist:

- Traditional RNNs, such as GRUs or LSTMs: these propagate information by keeping it in their hidden state from one timestep to the next. This temporally-linear dependency significantly inhibits the extent to which they can model long-range relationships in the data.

- Causal convolutions (van den Oord et al., 2016a; Salimans et al., 2017): these apply convolutions over the sequence (masked or shifted so that the current prediction is only influenced by previous element). They offer high-bandwidth access to the earlier parts of the sequence. However, their receptive field has a finite size, and still experience noticeable attenuation with regards to elements far away in the sequence.

- Self-attention (Vaswani et al., 2017): these models turn the sequence into an unordered key-value store that can be queried based on content. They feature an unbounded receptive field and allow undeteriorated access to information far away in the sequence. However, they only offer pinpoint access to small amounts of information, and require additional mechanism to incorporate positional information.

Causal convolutions and self-attention demonstrate complementary strengths and weaknesses: the former allow high bandwidth access over a finite context size, and the latter allow access over an infinitely large context. Interleaving the two thus offers the best of both worlds, where the model can have high-bandwidth access without constraints on the amount of information it can effectively use. The convolutions can be seen as aggregating information to build the context over which to perform an attentive lookup. Using this approach (dubbed SNAIL), Mishra et al. (2017) demonstrated significant performance improvements on a number of tasks in meta-learning setup, where the challenge of long-term temporal dependencies is also prevalent, as an agent should be able to adapt its behavior based on past experience.

In this note, we simply apply the same idea to the task of autoregressive generative modeling, as the fundamental bottleneck of access to past information is the same. Building off the current state-of-the-art in generative models, a class of convolution-based architectures known as PixelCNNs (van den Oord et al. (2016a) and Salimans et al. (2017)), we present a new architecture, PixelSNAIL, that incorporates ideas from (Mishra et al., 2017) to obtain state-of-the-art results on the CIFAR-10 and Imagenet $32 \times 32$ datasets.

## 2 MODEL ARCHITECTURE

In this section, we describe the PixelSNAIL architecture. It is primarily composed of two building blocks, which are illustrated in Figure 3 in Appendix and described below:

- A *residual block* applies several 2D-convolutions to its input, each with residual connections. To make them causal, the convolutions are masked so that the current pixel can only access pixels to the left and above from it. We use a gated activation function similar to (van den Oord et al., 2016a; Oord et al., 2016a). Throughout the model, we used 4 convolutions per block and 256 filters in each convolution.

- An *attention block* performs a single key-value lookup. It projects the input to a lower dimensionality to produce the keys and values and then uses softmax-attention like in (Vaswani et al., 2017; Mishra et al., 2017) (masked so that the current pixel can only attend over previously generated pixels). We used keys of size 16 and values of size 128.

Figure 4 (in Appendix) illustrates the full PixelSNAIL architecture, which interleaves the residual blocks and attention blocks depicted in Figure 3. In the CIFAR-10 model only, we applied dropout of 0.5 after the first convolution in every residual block, to prevent overfitting. We did not use any dropout for ImageNet, as the dataset is much larger. On both datasets, we use Polyak averaging Polyak & Juditsky (1992) (following (Salimans et al., 2017)) over the training parameters. We used an exponential moving average weight of 0.9995 for CIFAR-10 and 0.9997 for ImageNet. As the output distribution, we use the discretized mixture of logistics introduced by Salimans et al. (2017), with 10 mixture components for CIFAR-10 and 32 for ImageNet. To predict the subpixel (red,green,blue) values, we used the same linear-autoregressive parametrization as Salimans et al. (2017).

Our code is publicly available, and can be found at: `https://github.com/neocxi/pixelsnail-public`.

## 3 EXPERIMENTS

In Table 3, we provide negative log-likelihood results (in bits per dim) for PixelSNAIL on both CIFAR-10 and Imagenet $32 \times 32$. We compare PixelSNAIL's performance to a number of autoregressive models. These include: (i) PixelRNN (Oord et al., 2016b), which uses LSTMs, (ii) PixelCNN (van den Oord et al., 2016a) and PixelCNN++ (Salimans et al., 2017), which only use causal convolutions, and (iii) Image Transformer (Anonymous, 2018), an attention-only architecture inspired by Vaswani et al. (2017). PixelSNAIL outperforms all of these approaches, which suggests that both causal convolutions and attention are essential components of the architecture.

Table 1: Average negative log-likelihoods on CIFAR-10 and ImageNet $32 \times 32$, in bits per dim. PixelSNAIL outperforms other autoregressive models which only rely on causal convolutions xor self-attention.

| Method | CIFAR-10 | ImageNet $32 \times 32$ | ImageNet $64 \times 64$ |
|---|---|---|---|
| Conv DRAW (Gregor et al., 2016) | 3.5 | 4.40 | 4.10 |
| Real NVP (Dinh et al., 2016) | 3.49 | 4.28 | 3.98 |
| VAE with IAF (Kingma et al., 2016) | 3.11 | – | – |
| PixelRNN (Oord et al., 2016b) | 3.00 | 3.86 | 3.63 |
| Gated PixelCNN (van den Oord et al., 2016a) | 3.03 | 3.83 | 3.57 |
| Image Transformer (Anonymous, 2018) | 2.98 | 3.81 | – |
| PixelCNN++ (Salimans et al., 2017) | 2.92 | – | – |
| Block Sparse PixelCNN++ (OpenAI, 2017) | 2.90 | – | – |
| PixelSNAIL (ours) | **2.85** | **3.80** | **3.52** |

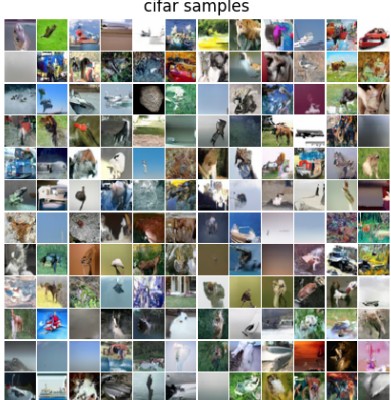
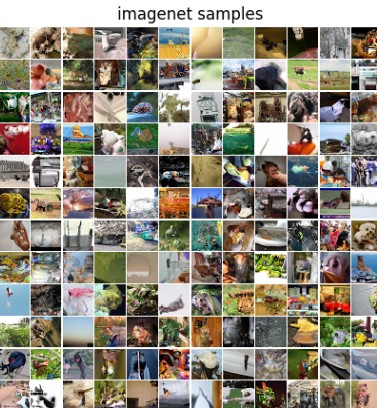

Figure 1: Samples from our CIFAR-10 model.   Figure 2: Samples from our ImageNet model.

## 4    CONCLUSION

We introduced PixelSNAIL, a class of autoregressive generative models that combine causal convolutions with self-attention. We demonstrate state-of-the-art density estimation performance on CIFAR-10 and ImageNet $32 \times 32$, with a publicly-available implementation at `https://github.com/neocxi/pixelsnail-public`.

Despite their tractable likelihood and strong empirical performance, one notable drawback of autoregressive generative models is that sampling is slow, because each pixel must be sampled sequentially. PixelSNAIL's sampling speed is comparable to that of existing autoregressive models; the design of models that allow faster sampling without losing performance remains an open problem.

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

APPENDIX

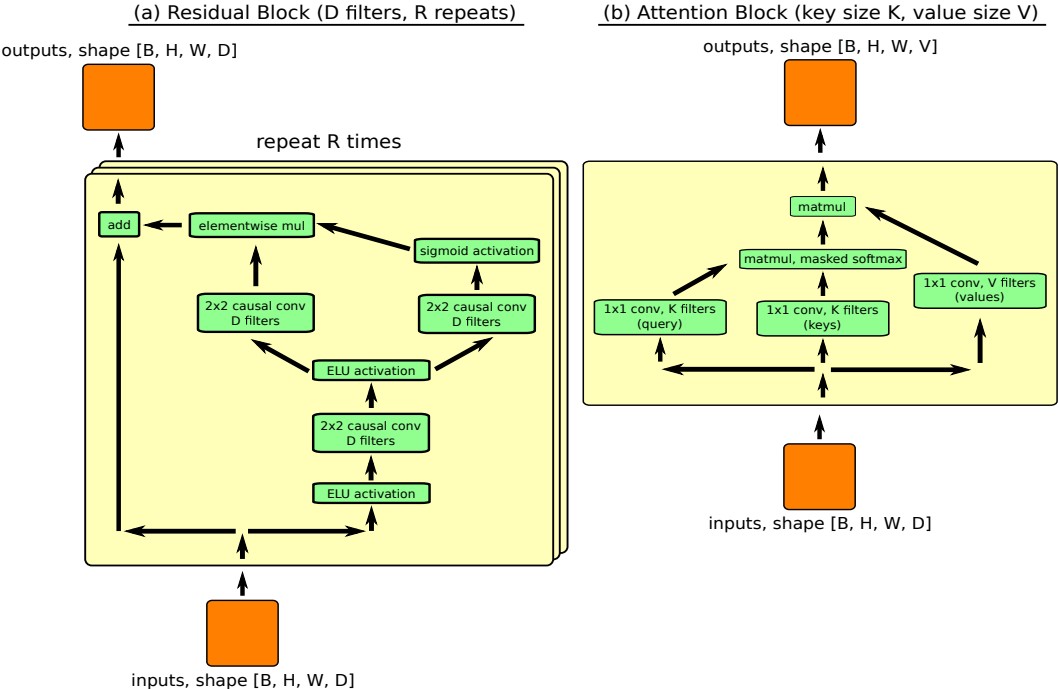

Figure 3:   The modular components that make up PixelSNAIL: (a) a residual block, and (b) an attention block. For both datasets, we used residual blocks with 256 filters and 4 repeats, and attention blocks with key size 16 and value size 128.

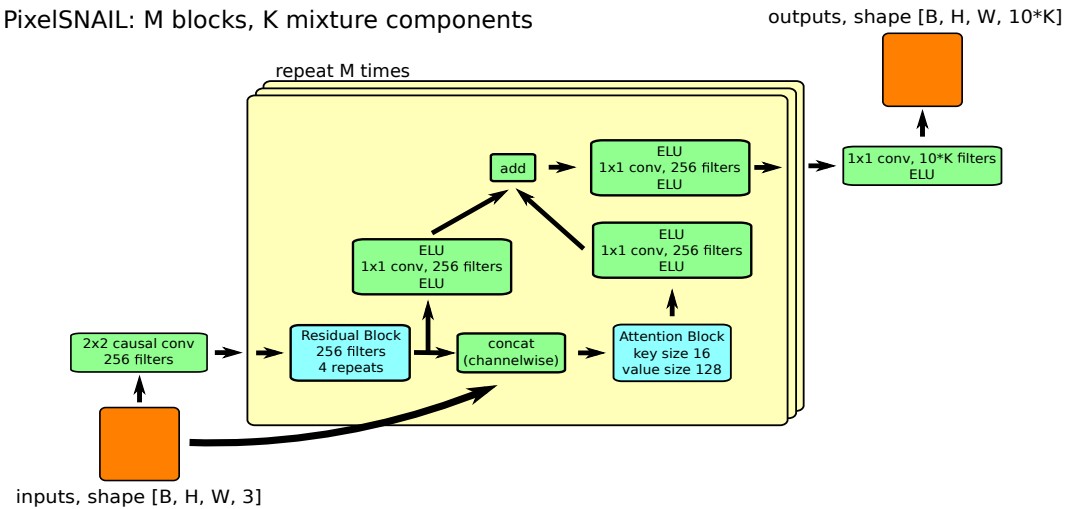

Figure 4:   The entire PixelSNAIL model architecture, using the building blocks from Figure 3. We used 12 blocks for both datasets, with 10 mixture components for CIFAR-10 and 32 for ImageNet.

