# OpenReview forum: "PixelSNAIL: An Improved Autoregressive Generative Model"
_ICLR.cc/2018/Workshop — Accept_

### Official Review · AnonReviewer2 · 2018-03-04
**Good paper**

**Rating:** 7
**Confidence:** 5

**Review:**

Summary:
This paper proposes to extend PixelCNN++ with attention blocks (Vaswani, 2017). The authors find that their model achieves state-of-the-art log-likelihoods on CIFAR-10 and ImageNet.

Review:
Exploring different architectures and pushing the limits of autoregressive models is a valuable contribution to the field of generative image modeling. I would have liked to read one or two more sentences comparing the proposed architecture to the Image Transformer (Vaswani, 2018), which also uses attention blocks to extend PixelCNNs, but achieves a more modest improvement.

---

### Official Review · AnonReviewer1 · 2018-03-06
**state-of-the-art log-likelihood on image benchmarks by adding attention to pixelRNN-style models**

**Rating:** 7
**Confidence:** 3

**Review:**

This paper presents an improved gate (building-block) for a pixelRNN type model. The improved gate combines an attention mechanism with causal convolution to obtain both high-bandwidth AND long-term memory. Experiments on standard benchmarks show an improvement across the board in terms of NLL, measuring how well the model fits the test data.
Pro: a clear improvement on the state-of-the-art for density estimation models
Cons: in a wider context, it is not clear to me how significant this improvement is - i.e. how does it translate into real applications - is going from 2.9 to 2.85 necessary to open the door to new applications ?

---

### Decision · Program_Chairs · 2018-03-20
**ICLR 2018 Workshop Acceptance Decision**

**Decision:**

Accept

**Comment:**

Congratulations, your paper was accepted to the ICLR workshop.